# OPEN SET RECOGNITION WITH GENERATED DATA

**Lawrence Neal, Matthew Olson, Weng-Keen Wong, Xiaoli Fern, & Fuxin Li**
Department of Electrical Engineering & Computer Science
Oregon State University
Corvallis, OR 97330, USA
`{nealla,olsomatt,wongwe,xfern,lif}@oregonstate.edu`

## ABSTRACT

In *open set recognition* tasks, a classifier must label instances of known classes while detecting unknown classes not encountered during training. We propose a framework in which a classifier jointly learns to classify instances of known classes, and to detect unknown classes as an additional "open set" class. Training examples for this open set class are synthesized by a generative adversarial network, itself trained only on the known classes. Augmenting the dataset with synthesized open set examples improves upon confidence thresholding.

## 1 INTRODUCTION

In the traditional image recognition task, all input is partitioned into a finite set of known classes, with equivalent training and testing distributions. However, many practical classification tasks may involve testing on *"unknown unknown"* classes not present in the training dataset (Scheirer et al., 2014), a situation that we refer to as *open set* image recognition. Typical methods for dealing with unknown classes involve thresholding the output confidence scores of a classifier. However, convolutional networks can output incorrect high-confidence predictions when faced with test data from outside the training distribution, as evidenced by work in adversarial example generation (Szegedy et al., 2013). We therefore focus on the active detection of open set examples as a class separate from the known training classes.

A number of models and training procedures have been proposed to make image recognition models robust to the open set of unknown classes. In (Jain et al., 2014), a support vector machine models the posterior probability of inclusion in each known class. In (Bendale & Boult, 2016), the distribution of the activations before the softmax layer of a classifier is examined to compute the probability of inclusion in any known class. A generative adversarial network is used in Schlegl et al. (2017) to compute a measure of probability of inclusion in a known set at test time, by mapping input images to points in the latent space of a generator. We propose a method complementary to these approaches, that requires no test-time computation apart from a prediction for a single additional class.

### 1.1 GENERATIVE MODELS AND OPEN SET RECOGNITION

We propose a type of dataset augmentation based on a generative model. The use of generative adversarial networks for data augmentation has been explored in eg. (Sixt et al., 2016), and generated examples were used as part of an open set training procedure in Ge et al. (2017). Instances of unknown classes are likely to have similar structure to known data, so a generative model trained on known classes is likely to learn features common to unknown classes.

However, supporting the observations of (Salimans et al., 2016) and (Dai et al., 2017), we find that visually realistic output is incompatible with strong unsupervised learning. A GAN that generates only realistic outputs will necessarily undergo mode collapse; only a "bad" GAN can capture all the variety of a training distribution. Based on this observation, we train a variant of adversarial autoencoder instead of a standard GAN. By strongly regularizing the learning process to avoid mode collapse, we can deliberately under-fit the distribution of training data. This allows our generative model to produce outputs that are outside the space of known classes, but still close enough to act as effective negative examples.

## 2 OPEN SET IMAGE RECOGNITION

We assume that a labeled training set $X$ consists of labeled examples of $K$ classes, and a test set contains $N > K$ classes, including the known classes in addition to one or more unknown classes. We pose the open set recognition problem as a classification among $K + 1$ classes, in which all instances of the $N - K$ unknown classes must be assigned to the additional class. Our objective is to first train a generative model capable of synthesizing simulated open set examples, and then to train a standard image classifier on a dataset containing the additional open set class.

The standard DCGAN training objective penalizes the generation of any images outside of the training distribution, and generators normally suffer from some level of mode collapse. Because this work aims to generate images outside the distribution of any training example, we require an output distribution that extends beyond any training examples. Inspired by the use of reconstruction losses to regularize the training of generators in Berthelot et al. (2017) and in Zhu et al., we use a training objective based on a combination of adversarial and reconstruction loss.

### 2.1 GENERATIVE MODEL

We train an encoder network $E$ which maps from images to a latent space, and a generator network $G$ which maps from latent space back to an image. The two networks are trained jointly as an autoencoder, with the objective to minimize the reconstruction error $||x - G(E(x))||$. Simultaneously, a discriminator network $D$ is trained as a Wasserstein critic with gradient penalty. The critic maximizes output for fake (autoencoded) images while minimizing output on real images. Training proceeds with alternating steps of optimization of the losses $L_D$ and $L_G$, where:

$$\mathbf{L_D} = \sum_{x \in \mathbf{X}} D(G(E(x))) - D(x) + P(D) \tag{1}$$

$$\mathbf{L_G} = \sum_{x \in \mathbf{X}} ||x - G(E(x))||_1 - D(G(E(x))) \tag{2}$$

where $P(D) = \lambda(||\nabla_{\hat{x}} D(\hat{x})||_2 - 1)$ is the interpolated gradient penalty term of (Gulrajani et al., 2017). Along with the generative model, we train a simple $K$-class classifier $C_K$ with cross-entropy loss on the labeled known classes.

### 2.2 SYNTHETIC OPEN SET EXAMPLES

An ideal open set data augmentation procedure will generate new examples that lie close to the original examples in a latent space, but on the other side of the true decision boundary between any known class and the open set. We formalize the notion of closeness as distance in the latent space $||z - E(x)||_2$ using the representation learned by $E$. To find a point outside the decision boundary of known classes, we perform an optimization by gradient descent in latent space, maximizing the entropy of output of the classifier $C_K$.

$$z^* = \min_z ||z - E(x)||_2^2 + \log\left(\sum_{i=0}^{K} \exp C_K(G(z))\right) \tag{3}$$

Given any $x$ we generate a latent point $z^*$ whose decoded representation $G(z^*)$ is augmented to the dataset with class label $K + 1$. After a sufficient number of open set examples have been synthesized, a new classifier $C_{K+1}$ is trained on the augmented dataset.

In the ablation experiment of section 3, we replace the latent space optimization with a random sampling (see 3). A model trained on sampled points can be shown to have the compact abating property of Scheirer et al. (2014) with respect to the latent space. However, the optimization from real examples towards regions of low confidence for $C_K$ provides better examples in practice.

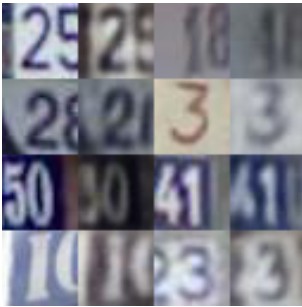 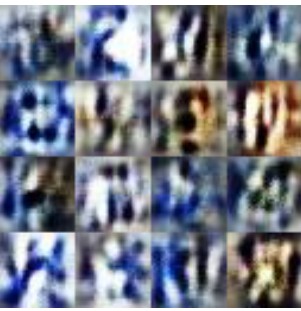 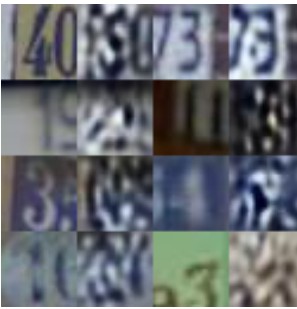

Figure 1: Left: Input/Output pairs of SVHN digits reconstructed by the autoencoder. Center: Examples generated from randomly-sampled points in the latent space. Right: Input/Output pairs of images generated from the latent space optimization procedure.

# 3 EXPERIMENTS

We evaluate the performance of the open set classifier $C_{K+1}$ by partitioning the classes of labeled datasets into known and unknown sets. At training time, the only input to the network consists of the $K$ known classes. At test time, the network must assign labels to the full set of $N$ classes. We evaluate using a subset of the the SVHN digit dataset, with $K = 5$, including only digits 0 through 4 in the training set and treating and 5 through 9 as unknown. We also test each of ten $K = 9$ subsets of MNIST.

To disentangle the objectives of classification and open set recognition, we evaluate each network with two metrics. To measure classification among known classes, we list the $K$-class classification accuracy. To measure the network's performance at discriminating between known and unknown inputs in a way invariant to the ratio of known to unknown examples, we treat the labeling of unknown classes as a detection task. A sensitivity threshold $\theta$ is applied to the open-set output of the classifier. The threshold $\theta$ is varied from perfect precision to perfect recall to generate an ROC curve, and the Area Under the Curve is calculated. For $C_{K+1}$ we apply this threshold to the softmax output for the open set class.

## 3.1 BASELINE AND ABLATION

We compare our open-set classification approach to a standard confidence-based method for the detection of unknown classes without dataset augmentation. The baseline network $C_K$ is trained only on known classes. Predictions such that $\max C_K(x) < \theta$ are counted as open set detections. We refer to this method as **Softmax Threshold**.

To test the effectiveness of the latent space optimization, we evaluate a version of the $C_{K+1}$ classifier with equation (3) removed. Instead, a point $z$ is drawn at random from the normal distribution with zero mean and unit variance. We refer to this method as **Generative-Random**. Finally the full method including equation (3) is **Generative-Optim**. Metrics reported for MNIST are the mean of ten runs.

|  | SVHN | | MNIST | |
| --- | --- | --- | --- | --- |
| Method | K-Class Accuracy | ROC AUC | K-Class Accuracy | ROC AUC |
| Softmax Threshold | .958 | .849 | .993 | .867 |
| Generative-Random | .951 | .886 | .989 | .946 |
| Generative-Optim. | .951 | **.900** | .993 | **.974** |

Figure 2: Classification accuracy for known classes and area under the ROC curve for unknown class detection in the SVHN dataset. In each experiment, $N = 10$.

ACKNOWLEDGMENTS

This material is based upon work supported by the Defence Advanced Research Projects Agency (DARPA) under contract N66001-17-2-4030.

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

Jun-Yan Zhu, Taesung Park, Phillip Isola, and Alexei A Efros. Unpaired image-to-image translation using cycle-consistent adversarial networks.

