# OpenReview forum: "Open Set Recognition with Generated Data"
_ICLR.cc/2018/Workshop — Reject_

### Official Review · AnonReviewer3 · 2018-03-10
**Complex procedure for limited results. Experiments incomplete and non convincing.**

**Rating:** 3
**Confidence:** 4

**Review:**

The paper considers the problem of multiclass classification where some examples do not belong to any of the known classes and shall be classified as unknown. It proposes to generate fake (unknown class) examples using a two steps procedure: first a GAN is trained together with a classifier (C_K) using the data with the known classes, then this GAN is used to learn “near the frontiers” examples. A new classifier (C_K+1) is then trained using both the labeled and the fake examples. Evaluation is performed on two image datasets.

This seems to me as an over complex procedure for dealing with the problem of unknown classes. The experimental section lacks precision. There is no indication on how the decision threshold theta is set for computing the accuracy on the softmax baseline. You do not indicate how many classes are supposed to be known in MNIST. There is no comparison with an alternative baseline making use of noisy data. Instead of training an alternative baseline with random samples, you could have sampled using the uncertainty provided by the classifier (e.g. sampling data for which no clear decision appears).
There is no indication about the nature of the GAN and of the classifiers components. Entropy is mentioned in Eq. (3) but the term used here is not an entropy. There is no indication on how the GAN and the classifier are trained (section 2.1), are you using an alternate optimization?

---

### Official Review · AnonReviewer2 · 2018-03-11
**Interesting technique using DCGAN for Data Augmentation**

**Rating:** 6
**Confidence:** 1

**Review:**

Clear illustration of technique but unsure about significance.

---

### Decision · Program_Chairs · 2018-03-20
**ICLR 2018 Workshop Acceptance Decision**

**Decision:**

Reject

**Comment:**

Based on the reviews, this paper has not been accepted for presentation at the ICLR workshop. However, the conversation and updates can continue to appear here on OpenReview.